# Awakening Visual Reasoning: Mitigating Post-Training Failure in Vision-Text Compression

**Xing Xi** [* 1] **Liyao Li** [* 2] **Hao Chen** [2] **Ningtao Wang** [3] **Peixian Chen** [3] **Peilin Tong** [3] **Xing Fu** [3] **Yu Cheng** [3] **Haobo Wang** [2] **Gang Chen** [2] **Junbo Zhao** [2] **Ronghua Luo** [1]

## Abstract

Vision-Text Compression (VTC) offers a scalable path for long-context multimodal modeling by rendering textual data into dense visual tokens. While recent Vision-Language Models (VLMs) demonstrate high decoding fidelity (OCR) on such inputs, they exhibit a severe *reasoning gap*: models that reason robustly on native text often fail on visually compressed equivalents, particularly in long-range retrieval and multi-step deduction. We identify a phenomenon of *post-training transfer failure*, where standard supervised fine-tuning and reinforcement learning on visual prompts yield marginal gains compared to their textual counterparts. To address this, we propose **CoRe** (**Co**ordinated **Re**asoning), a training framework that enforces *lockstep consistency* between the reasoning processes of textual and visual modalities. By treating the text-conditioned policy as a dynamic anchor, CoRe aligns the visual-conditioned policy via step-wise distribution matching, seamlessly integrating into both SFT and RL pipelines. Extensive evaluations across mathematical reasoning, long-context memory, and tabular retrieval benchmarks show that CoRe significantly outperforms standard visual post-training, recovering up to 70% of the performance gap relative to the textual upper bound and effectively activating latent reasoning capabilities in the compressed visual modality.

---

[*]Equal contribution [1]School of Computer Science and Engineering, South China University of Technology, Guangzhou, China [2]College of Computer Science and Technology, Zhejiang University, Hangzhou, China [3]Ant Group, Hangzhou, China. Correspondence to: Ningtao Wang <ningtao.nt@antgroup.com>, Haobo Wang <wanghaobo@zju.edu.cn>.

*Proceedings of the 43 $^{rd}$ International Conference on Machine Learning*, Seoul, South Korea. PMLR 306, 2026. Copyright 2026 by the author(s).

## 1. Introduction

Scaling large language models (LLMs) to increasingly long contexts is becoming essential for document understanding (Beltagy et al., 2020), code analysis (Feng et al., 2020), and complex multi-hop reasoning (Yang et al., 2018), yet remains computationally expensive due to the quadratic cost of attention and the linear growth of prefilling tokens (Vaswani et al., 2017). To mitigate this, *vision–text compression* (VTC) has emerged as a compelling paradigm (Wei et al., 2025b; Cheng et al., 2025): it renders long textual prompts into dense *text-as-image* inputs for vision–language models (VLMs), trading sparse token sequences for compact pixel grids to reduce text token budgets. Pioneering systems such as DeepSeek-OCR (Wei et al., 2025b) and Glyph (Cheng et al., 2025) show that, under carefully chosen rendering configurations, VTC can deliver multi-fold reductions in token usage while maintaining high decoding fidelity, positioning *visual context scaling* as a viable alternative to purely token-based long-context approaches.

Despite this progress, emerging evidence points to a persistent decoupling between *decoding* and *reasoning* under VTC. When the same semantic content is presented as a visually equivalent *visual prompt*, performance drops sharply on reasoning tasks requiring long-range dependency tracking, latent association, or multi-hop aggregation—even when the rendered text can be decoded accurately (Zhao et al., 2025; Clavié & Brand, 2025; Lyu et al., 2025). In other words, models that reason robustly from standard textual prompts often fail to exhibit comparable competence when operating on the corresponding visual prompts. Taken together, these results are most consistent with a failure to reliably couple decoded visual evidence with downstream reasoning, rather than a pure limitation of decoding fidelity.

A natural response is to apply post-training recipes that reliably enhance textual reasoning—supervised fine-tuning (SFT) on chain-of-thought data and reinforcement learning with verifiable rewards (RLVR)—directly to visual prompts. However, we find this strategy largely *ineffective* in the VTC setting. Across long-context capability axes where VTC is most attractive—*reasoning*, *memory*, and *retrieval*—SFT and RLVR yield clear gains with textual prompts but only

marginal improvements when the *same* prompts are presented as images (see Figure 1a). We term this phenomenon ***post-training transfer failure under VTC***. Concretely, under matched data and optimization budgets, post-training improvements learned from textual supervision transfer weakly to visual prompts, indicating a training-time misalignment in how reasoning improvements are invoked from rendered inputs.

Motivated by the principle of *predictive invariance*—since a textual prompt $P_t$ and its text-as-image rendering $P_v = \mathrm{Render}(P_t)$ convey identical semantic information, a competent model should induce identical reasoning policies under both views—we propose **CoRe** (**Co**ordinated **Re**asoning). CoRe addresses the transfer failure by introducing a plug-in consistency objective that keeps the reasoning distributions under visual prompts *in lockstep* with those under textual prompts, as illustrated in Figure 1b. Rather than relying solely on sparse outcome supervision (e.g., final answer correctness), CoRe provides **dense, process-level supervision**. Specifically, it adds an auxiliary KL-based alignment term $\mathcal{L}_{\mathrm{CoRe}}$ that encourages the visual branch to mimic the next-token probability distribution of the stronger text branch at every step of the reasoning chain.

Crucially, CoRe integrates seamlessly into standard post-training pipelines: (i) In *Supervised Fine-Tuning (SFT)*, it aligns visual-prompt predictions to ground-truth traces, ensuring the visual encoder triggers the correct reasoning path; (ii) In *Reinforcement Learning (RLVR)*, it leverages high-quality rollouts sampled from the text policy as *implicit reasoning targets*. This enables the visual branch to distill the model's own textual reasoning capabilities ("self-imitation") without requiring additional human annotation. Empirically, CoRe substantially narrows the reasoning gap under VTC (Figure 1c). On mathematical reasoning benchmarks, our method outperforms direct visual post-training by over 20%, proving that the reasoning bottleneck in VTC is largely a failure of alignment rather than capacity.

Our contributions are threefold:

- **Phenomenon Identification:** We systematically characterize the reasoning gap in text-as-image prompts and identify *post-training transfer failure* as a key obstacle: standard visual SFT/RL yields limited improvements despite high decoding fidelity.

- **The CoRe Framework:** We introduce CoRe, a model-agnostic regularizer that enforces lockstep consistency for coordinated reasoning. We demonstrate its unified application in both SFT and RL phases to align reasoning behavior across modalities without altering the fixed VTC rendering setup.

- **Extensive Empirical Validation:** We evaluate across mathematical reasoning, long-text question answering,

and table question answering benchmarks. Results show that CoRe consistently outperforms standard visual baselines and effectively preserves reasoning capabilities in the compressed vision-text modality. Our experiments focus on the high-fidelity VTC regime under current pipelines; we view the observed transfer failure as a stable empirical pattern in this setting rather than a universal claim about all future VLMs.

## 2. Methodology

We introduce **CoRe** (**Co**ordinated **Re**asoning), a training framework illustrated in Figure 1, which addresses the reasoning gap under Vision-Text Compression (VTC) by aligning reasoning behavior across semantically equivalent textual and visual prompts. Our key insight is to leverage cross-modal distribution matching as dense process supervision: rather than merely training the visual branch to produce correct outputs, we guide it to reason in a manner consistent with the stronger text branch. We begin by formalizing the dual-view setup, present the core alignment objective, detail its integration into standard post-training paradigms, and provide theoretical justification connecting distribution alignment to reasoning error control.

### 2.1. Problem Setup: Reasoning Under Dual View

Let $\mathcal{X}$ denote the space of reasoning task instances, and $\mathcal{Y}$ the space of output token sequences representing reasoning traces (e.g., chain-of-thought (Wei et al., 2022)) and final answers. A vision-language model (VLM) parameterized by $\theta$ defines an autoregressive policy

$$\pi_\theta(y \mid P) = \prod_{t=1}^{|y|} \pi_\theta(y_t \mid P, y_{<t}), \qquad (1)$$

where $P$ is the input prompt and $y_{<t}$ represents the generation history.

For each reasoning instance $x \in \mathcal{X}$, VTC creates two prompt representations that convey identical semantic content. The **text prompt** $P_t(x)$ is a standard token sequence encoding the task. The **visual prompt** $P_v(x) = \mathrm{Render}(P_t(x); \rho)$ is a text-as-image rendering, where $\rho$ specifies rendering parameters (resolution, layout, font size) chosen to balance compression ratio and text recoverability. Following established VTC protocols (Cheng et al., 2025), $\rho$ is determined through offline search and remains fixed during post-training. The rendering function is designed to preserve information: text content can be accurately decoded from $P_v(x)$ through optical character recognition.

Despite information equivalence, models exhibit systematic performance degradation when reasoning from visual prompts. Even when the VLM accurately decodes text from $P_v(x)$, it often fails on tasks requiring long-range

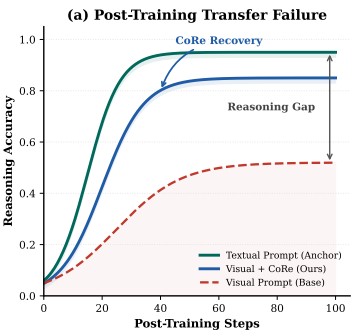 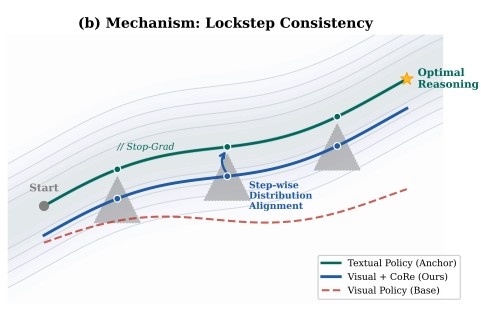 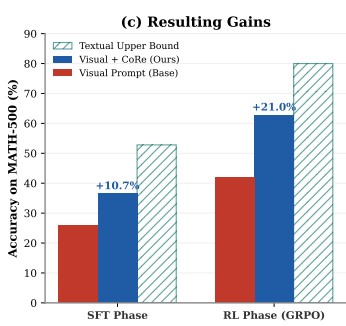

*Figure 1.* **Addressing the Post-Training Transfer Failure in Vision-Text Compression. (a) The Phenomenon:** Empirical training dynamics reveal a persistent reasoning gap. While the text policy (green) learns efficiently, the visual policy (red dashed) plateaus early despite identical semantic input, illustrating the *transfer failure*. CoRe (blue) successfully reactivates the visual reasoning circuit. **(b) The Mechanism:** We conceptualize reasoning as a trajectory through a latent probability manifold. The text policy serves as a stable anchor (green path) traversing the high-probability ridge. Standard visual post-training diverges into low-probability regions (red dashed). CoRe enforces *lockstep consistency* via step-wise KL projection (blue arrows), effectively "tethering" the visual reasoning process to the robust textual prior. **(c) Resulting Gains:** Quantitative comparison on MATH-500. CoRe significantly outperforms standard visual baselines, achieving a **+10.7%** gain in the SFT phase and a **+21.0%** gain in the RL phase, recovering a major portion of the performance gap relative to the textual upper bound (green hatched).

dependency tracking, multi-hop inference, or structured retrieval—tasks it handles well given $P_t(x)$. This suggests a failure mode beyond OCR limitations: the model struggles to reason over visually compressed content. We formalize this through a distributional lens. Let $\pi_\theta^t(\cdot \mid x) \triangleq \pi_\theta(\cdot \mid P_t(x))$ and $\pi_\theta^v(\cdot \mid x) \triangleq \pi_\theta(\cdot \mid P_v(x))$ denote output distributions under the two prompt types. The alignment gap is (Kullback & Leibler, 1951)

$$\Delta_{\text{align}}(\theta) = \mathbb{E}_{x \sim \mathcal{D}}\big[D_{\text{KL}}\big(\pi_\theta^t(\cdot \mid x) \,\big\|\, \pi_\theta^v(\cdot \mid x)\big)\big]. \quad (2)$$

Standard post-training approaches apply supervised fine-tuning or reinforcement learning directly to visual prompts, providing supervision on what final answer to produce but offering only sparse, outcome-level feedback. Empirically, we observe that improvements from text-prompt post-training transfer poorly to visual prompts—a phenomenon we term post-training transfer failure under VTC. This motivates our approach: explicitly aligning reasoning processes across modalities to enable knowledge transfer from the well-trained text channel to the weaker visual channel.

### 2.2. Core Method: Coordinated Reasoning Alignment

CoRe provides process-level supervision by matching the next-token probability distributions of the visual branch to those of the text branch along high-quality reasoning traces. This teaches the model not only what tokens to generate, but how to make reasoning decisions at each step.

Given an instance $x$ and a target reasoning trace $\tau = (\tau_1, \ldots, \tau_T)$, we align the two branches at each generation step. For $t \in \{1, \ldots, T\}$, define the next-token distributions conditioned on the common prefix $\tau_{<t}$:

$$\begin{aligned} q_\theta^{(t)} &= \pi_\theta(\cdot \mid P_t(x), \tau_{<t}), \\ p_\theta^{(t)} &= \pi_\theta(\cdot \mid P_v(x), \tau_{<t}). \end{aligned} \quad (3)$$

The instance-level CoRe alignment loss is the average step-wise KL divergence along the trace (Kullback & Leibler, 1951):

$$\mathcal{L}_{CoRe}(\theta; x, \tau) = \frac{1}{T}\sum_{t=1}^{T} D_{\text{KL}}\big(\text{sg}[q_\theta^{(t)}] \,\big\|\, p_\theta^{(t)}\big), \quad (4)$$

where $\text{sg}[\cdot]$ denotes the stop-gradient operator, which blocks gradients from flowing into $q_\theta^{(t)}$.

The distributions $q_\theta^{(t)}$ and $p_\theta^{(t)}$ encode the reasoning policies of the text and visual branches—they specify what next token to generate given the current reasoning state. By minimizing their KL divergence with stop-gradient, we treat the text branch as a reference and update only the visual branch to make similar reasoning decisions at each step. This distribution-matching view is closely related to knowledge distillation (Hinton et al., 2015). This is richer than standard supervised learning, which only enforces high probability on the correct token $\tau_t$ without constraining the distribution over alternatives. Distribution matching transfers the entire decision-making strategy, including the model's uncertainty and preference structure over plausible continuations.

During training, both branches are conditioned on the same prefix $\tau_{<t}$ to create a shared reference frame for comparison. At each step, we compare their reasoning strategies (probability distributions) rather than just final outputs. While

at test time the visual branch generates its own trajectory independently, the training procedure shapes its learned policy—the way it assigns probabilities at each decision point—to mirror the text branch's reasoning patterns. By training over diverse reasoning traces, the visual branch learns a generalizable reasoning strategy rather than memorizing specific paths.

In general, we may have multiple candidate traces per instance, each with an associated weight. We define a weighted trace set $\mathcal{S}(x) = \{(\tau^{(k)}, w^{(k)})\}_{k=1}^{K_x}$ where $w^{(k)} \geq 0$ and $\sum_k w^{(k)} = 1$. The weighted alignment objective is

$$\bar{\mathcal{L}}_{CoRe}(\theta; x, \mathcal{S}(x)) = \sum_{k=1}^{K_x} w^{(k)} \, \mathcal{L}_{CoRe}(\theta; x, \tau^{(k)}). \quad (5)$$

This formulation unifies supervised and reinforcement learning settings, which differ only in how $\mathcal{S}(x)$ is constructed.

### 2.3. Integration into Post-Training Pipelines

CoRe serves as a modular regularization term that augments standard training objectives:

$$\mathcal{L}_{\text{total}}(\theta) = \mathcal{L}_{\text{base}}(\theta) \\ + \lambda \cdot \mathbb{E}_{x \sim \mathcal{D}}\big[\bar{\mathcal{L}}_{CoRe}(\theta; x, \mathcal{S}(x))\big], \quad (6)$$

where $\mathcal{L}_{\text{base}}$ is the task-specific training loss, $\lambda > 0$ controls alignment strength, and $\mathcal{D}$ is the training distribution.

**Supervised fine-tuning.** In SFT, each training instance $(x, \tau^\star)$ includes a ground-truth reasoning trace. The base objective applies standard next-token prediction to both prompt types:

$$\mathcal{L}_{\text{SFT}}(\theta; x, \tau^\star) = -\log \pi_\theta(\tau^\star \mid P_t(x)) \\ - \log \pi_\theta(\tau^\star \mid P_v(x)). \quad (7)$$

The trace set is simply $\mathcal{S}(x) = \{(\tau^\star, 1)\}$. The full SFT+CoRe objective becomes:

$$\mathcal{L}_{\text{SFT+CoRe}}(\theta) = \mathbb{E}_{(x,\tau^\star) \sim \mathcal{D}_{\text{sft}}}\big[\mathcal{L}_{\text{SFT}} \\ + \lambda \cdot \mathcal{L}_{CoRe}(\theta; x, \tau^\star)\big]. \quad (8)$$

Here, the base loss provides outcome supervision (correct tokens to generate), while CoRe provides process supervision (how to reason).

**Reinforcement learning.** In RLVR, we lack ground-truth traces but have a reward function $R(\cdot; x)$ verifying answer correctness. We leverage the text branch to generate candidate traces through sampling. For each instance $x$, we sample $K$ rollouts from the text-conditioned policy:

$$y^{(k)} \sim \pi_\theta(\cdot \mid P_t(x)), \\ r^{(k)} = R(y^{(k)}; x), \quad k = 1, \dots, K. \quad (9)$$

We then apply a selection or weighting scheme. A common approach assigns uniform weight to correct rollouts and zero weight to incorrect ones:

$$\omega^{(k)} = \mathbb{I}[r^{(k)} = 1], \\ w^{(k)} = \frac{\omega^{(k)}}{\sum_{j=1}^K \omega^{(j)}}. \quad (10)$$

This yields the trace set $\mathcal{S}(x) = \{(y^{(k)}, w^{(k)}) : \omega^{(k)} > 0\}$. When no rollout is correct, i.e., $\sum_k \omega^{(k)} = 0$, we have $\mathcal{S}(x) = \emptyset$ and the CoRe term for instance $x$ is zero; only the base GRPO loss contributes in this case. We adopt Group Relative Policy Optimization (GRPO) (Shao et al., 2024) as our base RL algorithm, which is closely related to proximal policy optimization (PPO) (Schulman et al., 2017). The full RL+CoRe objective is:

$$\mathcal{L}_{\text{RL+CoRe}}(\theta) = \mathcal{L}_{\text{GRPO}}(\theta) \\ + \lambda \cdot \mathbb{E}_x \left[ \sum_{k:\omega^{(k)}>0} w^{(k)} \mathcal{L}_{CoRe}(\theta; x, y^{(k)}) \right]. \quad (11)$$

In this setting, CoRe enables self-improvement: as training progresses, the text branch generates increasingly higher-quality rollouts (due to GRPO optimization), which serve as better alignment targets for the visual branch. The reward-based filtering ensures we only align to correct reasoning paths, preventing error propagation.

### 2.4. Unified Training Algorithm

Algorithm 1 presents the complete CoRe training procedure in a unified framework that encompasses both SFT and RL settings through conditional branching.

The algorithm differs only in how the trace set $\mathcal{S}(x)$ and base loss are constructed for each instance. CoRe requires forward passes through both prompt types to compute distributions $q_\theta^{(t)}$ and $p_\theta^{(t)}$. In practice, the text-branch forward pass is already required for the base loss, and the visual-branch forward pass is shared between the base visual loss and CoRe. The main additional cost comes from computing KL divergences at each step.

### 2.5. Theoretical Justification

We now provide theoretical justification for why minimizing the cross-view KL divergence controls the reasoning error under visual prompts.

For a given instance $x$, let $\pi^\star(\cdot \mid x)$ denote an oracle policy that assigns high probability only to correct reasoning traces. We measure reasoning error using total variation

**Algorithm 1** Unified CoRe Training Framework

**Require:** Training data $\mathcal{D}$, initial VLM $\theta_0$, rendering config $\rho$, alignment weight $\lambda$, mode $\in \{\text{SFT}, \text{RL}\}$
1: Initialize $\theta \leftarrow \theta_0$
2: **for** each training iteration **do**
3:     Sample mini-batch $\mathcal{B}$ from $\mathcal{D}$
4:     Initialize: $\mathcal{L}_{\text{batch}}^{\text{base}} \leftarrow 0$, $\mathcal{L}_{\text{batch}}^{CoRe} \leftarrow 0$
5:     **for** each instance $x \in \mathcal{B}$ **do**
6:         Construct prompts: $P_t \leftarrow P_t(x)$, $P_v \leftarrow$ Render$(P_t; \rho)$
7:         **if** mode is SFT **then**
8:             $\mathcal{S}(x) \leftarrow \{(\tau^\star, 1.0)\}$ where $\tau^\star$ is ground-truth trace
9:             $\mathcal{L}_{\text{base}}^{(x)} \leftarrow -\log \pi_\theta(\tau^\star \mid P_t) - \log \pi_\theta(\tau^\star \mid P_v)$
10:         **else if** mode is RL **then**
11:             Sample: $\{y^{(k)}\}_{k=1}^K \sim \pi_\theta(\cdot \mid P_t)$
12:             Compute: $r^{(k)} \leftarrow R(y^{(k)}; x)$ for $k = 1, \ldots, K$
13:             Construct $\mathcal{S}(x)$ from $\{(y^{(k)}, r^{(k)})\}$ via Eq. (10)
                  $\triangleright$ $\mathcal{S}(x) = \emptyset$ if no correct rollout
14:             $\mathcal{L}_{\text{base}}^{(x)} \leftarrow \mathcal{L}_{\text{GRPO}}(\theta; x, \{y^{(k)}, r^{(k)}\})$
15:         **end if**
16:         Initialize $\mathcal{L}_{CoRe}^{(x)} \leftarrow 0$
17:         **for** each trace $(\tau, w) \in \mathcal{S}(x)$ **do**
18:             **for** $t = 1$ to $|\tau|$ **do**
19:                 Compute: $q_\theta^{(t)} \leftarrow \pi_\theta(\cdot \mid P_t, \tau_{<t})$, $p_\theta^{(t)} \leftarrow \pi_\theta(\cdot \mid P_v, \tau_{<t})$
20:                 $\mathcal{L}_{CoRe}^{(x)} \leftarrow \mathcal{L}_{CoRe}^{(x)} + w \cdot D_{\text{KL}}(\text{sg}[q_\theta^{(t)}] \| p_\theta^{(t)})/|\tau|$
21:             **end for**
22:         **end for**
23:         $\mathcal{L}_{\text{batch}}^{\text{base}} \leftarrow \mathcal{L}_{\text{batch}}^{\text{base}} + \mathcal{L}_{\text{base}}^{(x)}$
24:         $\mathcal{L}_{\text{batch}}^{CoRe} \leftarrow \mathcal{L}_{\text{batch}}^{CoRe} + \mathcal{L}_{CoRe}^{(x)}$
25:     **end for**
26:     $\mathcal{L} \leftarrow \frac{1}{|\mathcal{B}|}(\mathcal{L}_{\text{batch}}^{\text{base}} + \lambda \cdot \mathcal{L}_{\text{batch}}^{CoRe})$
27:     Update: $\theta \leftarrow \theta - \eta \nabla_\theta \mathcal{L}$
28: **end for**

(TV) distance (Cover & Thomas, 2006):

$$\epsilon(\pi; x) = \text{TV}(\pi(\cdot \mid x), \pi^\star(\cdot \mid x)) = \sup_{A \subseteq \mathcal{Y}} |\pi(A \mid x) - \pi^\star(A \mid x)|. \quad (12)$$

This captures the worst-case probability mass assigned to incorrect traces. Our goal is to bound the visual reasoning error $\epsilon(\pi_\theta^v; x)$ in terms of quantities we control through training.

**Proposition 2.1** (Upper bound on visual reasoning error). *Let $\pi_\theta^t(\cdot \mid x)$ and $\pi_\theta^v(\cdot \mid x)$ denote the text and visual policies. Define:*

$$\epsilon_t(\theta; x) = TV(\pi_\theta^t(\cdot \mid x), \pi^\star(\cdot \mid x)),$$
$$\epsilon_v(\theta; x) = TV(\pi_\theta^v(\cdot \mid x), \pi^\star(\cdot \mid x)). \quad (13)$$

*Then:*

$$\epsilon_v(\theta; x) \leq \epsilon_t(\theta; x) + \sqrt{\frac{1}{2} D_{KL}(\pi_\theta^t \| \pi_\theta^v)}, \quad (14)$$

*where we write $\pi_\theta^t$ and $\pi_\theta^v$ as shorthand for $\pi_\theta^t(\cdot \mid x)$ and $\pi_\theta^v(\cdot \mid x)$.*

*Proof.* By the triangle inequality of total variation distance:

$$\begin{aligned} \text{TV}(\pi_\theta^v, \pi^\star(\cdot \mid x)) \\ \leq \text{TV}(\pi_\theta^v, \pi_\theta^t) + \text{TV}(\pi_\theta^t, \pi^\star(\cdot \mid x)). \end{aligned} \quad (15)$$

The second term equals $\epsilon_t(\theta; x)$ by definition. Applying Pinsker's inequality (Cover & Thomas, 2006), $\text{TV}(P, Q) \leq \sqrt{\frac{1}{2} D_{\text{KL}}(P \| Q)}$, to the first term with $P = \pi_\theta^t$ and $Q = \pi_\theta^v$ yields the result. $\square$

This bound decomposes the visual reasoning error into two interpretable components. The term $\epsilon_t(\theta; x)$ represents how well the text branch reasons. For strong pretrained VLMs with text-focused post-training, this baseline error is typically small and stable. The second term grows with the KL divergence between the two policies. By minimizing $D_{\text{KL}}(\pi_\theta^t \| \pi_\theta^v)$, we directly tighten the upper bound on visual reasoning error.

While Proposition 2.1 involves the sequence-level KL divergence over complete output sequences, our practical objective in Eq. (4) minimizes step-wise KL divergences along specific traces. These are connected through the chain rule of KL divergence for autoregressive models (Cover & Thomas, 2006). For any trace $\tau = (\tau_1, \ldots, \tau_T)$, the autoregressive factorization gives:

$$\log \frac{\pi_\theta^t(\tau \mid x)}{\pi_\theta^v(\tau \mid x)} = \sum_{t=1}^T \log \frac{\pi_\theta(\tau_t \mid P_t, \tau_{<t})}{\pi_\theta(\tau_t \mid P_v, \tau_{<t})}. \quad (16)$$

Taking the expectation over $\tau \sim \pi_\theta^t$ and marginalizing over prefixes, the sequence-level KL decomposes as:

$$\begin{aligned} &D_{\text{KL}}(\pi_\theta^t \| \pi_\theta^v) \\ &= \sum_{t=1}^T \mathbb{E}_{\tau_{<t} \sim \pi_\theta^t} \Big[ D_{\text{KL}}(\pi_\theta(\cdot \mid P_t, \tau_{<t}) \| \pi_\theta(\cdot \mid P_v, \tau_{<t})) \Big], \end{aligned} \quad (17)$$

where each term is the expected step-wise KL conditioned on the prefix $\tau_{<t}$ drawn from the text policy. This is precisely our shorthand $D_{\text{KL}}(q_\theta^{(t)} \| p_\theta^{(t)})$ averaged over the prefix distribution.

The CoRe objective minimizes step-wise KL terms along a representative set of traces $\mathcal{S}(x)$. When these traces are drawn from or approximate the text policy's distribution (as in RL) or cover the support of correct reasoning paths (as in

SFT), minimizing $\mathcal{L}_{CoRe}$ effectively reduces the sequence-level divergence that appears in the theoretical bound. This bridges the gap between our practical training objective and the theoretical guarantee, showing that CoRe is a principled approach grounded in controlling distributional error.

## 3. Experiments

We evaluate CoRe across three cognitive axes critical for VTC: **Reasoning**, **Memory**, and **Retrieval**. Our experiments aim to validate whether lockstep alignment can bridge the performance gap between visual and textual prompting across these diverse capabilities, even in regimes lacking ground-truth reasoning traces.

### 3.1. Datasets and Evaluation Benchmarks

To isolate reasoning capabilities from perceptual variance, we employ **Qwen3-VL-8B-Instruct** (Bai et al., 2025) with a fixed rendering configuration $\rho$ following prior VTC protocols (Cheng et al., 2025), ensuring high OCR fidelity across all visual prompts. We evaluate all tasks using **Exact Match (Accuracy)** on the final answer. We ground our evaluation in three domains, each paired with specific training resources and benchmarks.

Importantly, the text branch that serves as the alignment anchor is the *same* model receiving the original textual prompt—not a separate, frozen backbone. This design choice is deliberate: because CoRe aligns the visual policy toward the text policy of the *same* checkpoint, both branches co-evolve during training, and the text branch's improvement through GRPO naturally raises the alignment ceiling for the visual branch.

**Mathematical Reasoning.** This domain corresponds to the *Reasoning* axis. For training, we utilize the **DeepScaleR** dataset (Luo et al., 2025), employing a subset of 7K samples with ground-truth Chain-of-Thought (CoT) for the SFT phase, and a larger subset of 32K samples without CoT for the RL phase. Unless otherwise specified, the RLVR experiments in the main results follow the standard GRPO setting with $K = 8$ rollouts per prompt. Evaluation is conducted on **MATH-500** (Hendrycks et al., 2021), **GSM8K** (Cobbe et al., 2021), **AMC 2023** (Mathematical Association of America, 2023), and **AIME 2024/2025** (Mathematical Association of America, 2025), assessing multi-step deductive logic.

**Long-Context Understanding.** Corresponding to the *Memory* axis, this domain challenges the model to retain and manipulate information over long visual sequences. Since human-annotated CoT is scarce here, we bypass SFT and apply CoRe-RL directly. We train on **DocQA-RL-1.6K** (Wan et al., 2025) and evaluate on a suite of benchmarks: **DocMath** (Zhao et al., 2024), **FRAME** (Krishna et al., 2025), **HotpotQA** (Yang et al., 2018), **2WikiMultihopQA** (Ho

et al., 2020), **MuSiQue** (Trivedi et al., 2022), **NarrativeQA** (Kočiský et al., 2018), and **Qasper** (Dasigi et al., 2021).

**Tabular QA.** Mapping to the *Retrieval* axis, this domain requires precise cell localization and aggregation. We construct a training mixture of 102K samples from the five following table QA datasets, randomly sampling 6K instances for the RL phase. Evaluation covers Question Answering tasks (**TabMWP** (Lu et al., 2023), **HiTab** (Cheng et al., 2022), **TAT-QA** (Zhu et al., 2021)) and Fact Verification tasks (**TabFact** (Chen et al., 2020), **InfoTabs** (Gupta et al., 2020)).

### 3.2. Main Results and Analysis

We present the performance of CoRe in Table 1 (Math), Table 2 (Long-Context), and Table 3 (Tabular QA).

**Mathematical Reasoning and the SFT Gap.** As shown in Table 1, the base model exhibits a profound performance disparity between modalities. While the text-prompted model achieves a strong average accuracy of 51.46% (0-shot) and improves to 63.54% with standard Text GRPO, the visually-prompted counterparts lag significantly behind, with standard Visual SFT averaging only 15.42%. This confirms that standard visual post-training fails to elicit high-level reasoning circuits. However, introducing CoRe alignment during SFT significantly mitigates this failure, improving the visual average to 20.92% and providing a superior initialization for subsequent RL.

**Closing the Gap via CoRe-RL.** The most substantial gains are observed when deploying the full CoRe-RL pipeline. In the mathematical domain, our method achieves a state-of-the-art visual average of 42.97%, recovering the majority of the reasoning capabilities possessed by the text upper bound. On GSM8K, CoRe reaches 82.19% compared to 67.31% for the visual baseline. Crucially, this trend extends to the Memory and Retrieval domains where we lack ground-truth CoT. In Long-Context understanding (Table 2), standard Visual GRPO yields marginal gains over zero-shot baselines (e.g., HotpotQA improves only from 7.0% to 9.0%), suggesting that sparse outcome-based rewards are insufficient to navigate complex visual contexts. In contrast, CoRe-RL effectively "activates" these capabilities, jumping to 23.0% on HotpotQA and 35.0% on 2WikiMultihopQA.

**Consistent Gains in Structured Retrieval.** Similar breakthroughs are seen in Tabular QA (Table 3). Standard Visual GRPO struggles to improve over the zero-shot baseline, particularly on complex tasks like HiTab (improving marginally from 0.6% to 3.2%). CoRe-RL, however, leverages the text model's superior ability to serialize and reason over table cells, boosting HiTab performance to 5.1% and achieving remarkable gains on InfoTabs (from 31.1% to 70.2%). This validates that even for structured data, aligning the visual

*Table 1.* **Mathematical Reasoning (Reasoning Axis).** Accuracy (%) under dual-view prompting. **V**: visual prompt ($P_v$). **T**: textual prompt ($P_t$). Gray rows indicate text-only upper bounds. CoRe successfully transfers reasoning patterns from text to vision, enabling the model to solve competition-level problems (AIME) visually.

| | MATH-500 | | GSM8K | | AMC | | AIME 24 | | AIME 25 | | Avg |
|---|---|---|---|---|---|---|---|---|---|---|---|
| **Method** | V | T | V | T | V | T | V | T | V | T | V / T |
| Zero-shot Inference | 14.80 | 70.40 | 7.79 | 92.68 | 6.02 | 54.22 | 0.00 | 16.67 | 3.33 | 23.33 | 6.39 / 51.46 |
| Text SFT (1 epoch) | – | 52.80 | – | 82.34 | – | 27.71 | – | 3.33 | – | 0.00 | – / 33.24 |
| $\rightarrow$ Text GRPO (Iterative) | – | 79.20 | – | 94.09 | – | 81.93 | – | 43.33 | – | 26.67 | – / 65.04 |
| Text GRPO (Directly on Base) | 25.00 | 80.00 | 30.98 | 94.20 | 7.23 | 73.49 | 0.00 | 43.33 | 3.33 | 26.67 | 13.31 / 63.54 |
| Visual SFT (1 epoch) | 25.80 | – | 41.97 | – | 6.02 | – | 0.00 | – | 3.33 | – | 15.42 / – |
| $\rightarrow$ Visual GRPO | 41.60 | – | 50.63 | – | 14.46 | – | 0.00 | – | 13.33 | – | 24.00 / – |
| Visual GRPO (Directly on Base) | 41.80 | – | 67.31 | – | 18.07 | – | 10.00 | – | 10.00 | – | 29.44 / – |
| CoRe-SFT (on Text SFT) | 36.00 | – | 36.54 | – | 12.05 | – | 0.00 | – | 20.00 | – | 20.92 / – |
| $\rightarrow$ Visual GRPO | 51.60 | – | 69.10 | – | 22.89 | – | 16.67 | – | 10.00 | – | 34.05 / – |
| CoRe-RL (on Zero-shot) | 37.60 | – | 49.49 | – | 18.07 | – | 3.33 | – | 10.00 | – | 23.70 / – |
| $\rightarrow$ Visual SFT | 57.00 | – | 74.01 | – | **28.92** | – | **26.67** | – | 16.67 | – | 40.65 / – |
| $\rightarrow$ **Visual GRPO (Final)** | **62.80** | 79.80 | **82.19** | 94.51 | 26.51 | 74.70 | 23.33 | 40.00 | **20.00** | 30.00 | **42.97** / 63.80 |

*Table 2.* **Long-Context Understanding (Memory Axis).** Performance comparison across visual and textual modalities. CoRe-RL significantly outperforms standard visual GRPO.

| | Visual Prompt | | | Textual Prompt | |
|---|---|---|---|---|---|
| **Benchmark** | Infer. | GRPO | **CoRe-RL** | Infer. | GRPO |
| DocMath | 6.14 | 6.50 | **14.50** | 56.00 | 60.00 |
| FRAME | 6.00 | 8.00 | **13.00** | 32.00 | 32.00 |
| HotpotQA | 7.00 | 9.00 | **23.00** | 45.00 | 51.00 |
| 2WikiQA | 10.00 | 13.00 | **35.00** | 54.00 | 65.00 |
| MuSiQue | 1.00 | 1.00 | **7.00** | 32.00 | 31.00 |
| NarrativeQA | 2.00 | 4.00 | **4.00** | 12.00 | 12.00 |
| Qasper | 0.00 | 1.00 | **4.00** | 16.00 | 15.00 |

*Table 3.* **Tabular QA (Retrieval Axis).** Evaluation on structured data benchmarks. CoRe-RL consistently improves over the visual baseline.

| | Visual Prompt | | | Textual Prompt | |
|---|---|---|---|---|---|
| **Benchmark** | Infer. | GRPO | **CoRe-RL** | Infer. | GRPO |
| HiTab | 0.63 | 3.16 | **5.06** | 35.44 | 0.00 |
| InfoTabs | 31.11 | 61.29 | **70.18** | 68.51 | 45.92 |
| TabFact | 24.81 | 51.97 | **61.31** | 72.11 | 31.53 |
| TabMWP | 24.83 | 53.83 | **61.11** | 68.40 | 42.52 |
| TAT-QA | 7.18 | 23.35 | **26.34** | 46.10 | 12.57 |

reasoning process with the textual process is more effective than reinforcement learning from scratch. Notably, Text GRPO itself drops to near-zero on HiTab (0.0%), suggesting that the reward signal for hierarchical tables is extremely sparse: when few rollouts yield a correct answer, GRPO alone cannot learn an effective policy, regardless of modality. CoRe mitigates this by providing a denser, process-level supervision signal that does not depend on outcome-level reward frequency.

### 3.3. Ablation Studies

To understand the optimal configuration for CoRe, we conduct a unified ablation study on the mathematical reasoning benchmarks. We investigate two critical dimensions: the source of the alignment signal (SFT vs. RL) and the scale of rollout sampling ($K$). The results are summarized in Table 4.

**Impact of Alignment Source (SFT vs. RL).** We first compare the effectiveness of aligning to a ground-truth trace

(CoRe-SFT) versus aligning to a single model-generated rollout verified by reward (CoRe-RL, $K = 1$). The results reveal a task-dependent preference. On challenging benchmarks like **AIME 2024** and **AIME 2025**, CoRe-SFT (using ground-truth CoT) outperforms CoRe-RL ($K = 1$) (e.g., **6.67%** vs. 3.33% on AIME 2024, **16.67%** vs. 10.00% on AIME 2025). We hypothesize that for such hard problems, the weak visual policy faces an exploration bottleneck and rarely generates correct solutions on its own; thus, the dense supervision from ground-truth traces is critical for injecting new reasoning knowledge. Conversely, on moderate tasks like **MATH-500** and **AMC**, CoRe-RL ($K = 1$) yields comparable or better performance (e.g., **42.80%** vs. 42.20% on MATH-500, **22.89%** vs. 20.48% on AMC). This suggests that when the model is capable of self-generating solutions, aligning to its *own* correct reasoning path—which perfectly matches its intrinsic distribution—may be more effective than forcing it to mimic an external human annotation distribution.

**Impact of Rollout Scale ($K$).** We further examine the effect of increasing the number of sampled rollouts $K$ in the

RL phase. As shown in Table 4 (Comparison 2), performance generally improves as $K$ increases from 1 to 4. For instance, MATH-500 accuracy rises from 42.80% ($K = 1$) to **47.20%** ($K = 4$), and GSM8K from 58.54% to **65.89%**. This trend suggests that aggregating gradients from multiple correct rollouts stabilizes the alignment signal. However, we observe a performance drop at $K = 8$ across several benchmarks (e.g., GSM8K drops to 54.81%). We attribute this to **Mode Averaging Conflict**: when aligning to a large number of diverse correct reasoning paths simultaneously, the model may be forced towards an "average" distribution that lacks the sharpness and coherence of any single successful strategy. This indicates that a moderate number of rollouts (e.g., $K = 4$) strikes an optimal balance between signal robustness and mode consistency.

*Table 4.* **Unified Ablation Study.** Impact of alignment source (SFT vs. RL) and rollout scale ($K$) on mathematical reasoning. SFT is more effective for hard tasks (AIME), while RL benefits moderate tasks. Performance generally peaks at $K = 4$, suggesting that excessive rollouts ($K = 8$) may introduce mode conflicts.

| Methods | MATH-500 | GSM8K | AMC | AIME 24 | AIME 25 |
|---|---|---|---|---|---|
| CoRe-SFT (Oracle) | 42.20 | 60.73 | 20.48 | 6.67 | **16.67** |
| CoRe-RL (Rollout $K = 1$) | **42.80** | 58.54 | **22.89** | 3.33 | 10.00 |
| CoRe-RL (Rollout $K = 2$) | 44.60 | 65.33 | 19.28 | 10.00 | 6.67 |
| **CoRe-RL (Rollout $K = 4$)** | **47.20** | **65.89** | **25.30** | **10.00** | 10.00 |
| CoRe-RL (Rollout $K = 8$) | 44.20 | 54.81 | 18.07 | **10.00** | 10.00 |

### 3.4. Failure Mode Analysis

To better understand the remaining performance gap, we manually inspect predictions in the hardest settings and identify two dominant failure patterns:

1. **Hierarchical-table localization errors.** On datasets such as HiTab, the model often reads the table correctly but produces malformed or off-by-one cell references when reasoning over multi-level structure. This explains the anomalously low HiTab scores even when OCR is accurate.

2. **Long-range dependency tracking failures.** On very long visual contexts, especially MuSiQue and Qasper, the model struggles to preserve multi-hop dependencies across distant rendered segments even when local OCR is correct.

Importantly, CoRe still provides meaningful gains in these hard regimes—for example, MuSiQue improves from 1.0 to 7.0, and Qasper from 0.0 to 4.0—suggesting that these are compression-induced reasoning bottlenecks rather than fundamental incapacities.

## 4. Related Work

**Vision–Text Compression and the Reasoning Gap.**
Vision–Text Compression (VTC) has emerged as a scalable paradigm for processing long contexts, rendering textual data into dense visual representations to bypass the quadratic complexity of attention mechanisms. Pioneering systems like DeepSeek-OCR (Wei et al., 2025b) and Glyph (Cheng et al., 2025) demonstrate that optimizing rendering configurations (e.g., resolution, DPI) allows VLMs to encode thousands of text tokens into fixed-size visual patches with high decodability. However, a growing body of evidence indicates a systematic "read–reason" decoupling. While earlier OCR-free models (Kim et al., 2022; Wei et al., 2025a) focused on perception, recent benchmarks reveal that VLMs struggle with higher-order cognition on compressed inputs. VTCBench (Zhao et al., 2025) identifies specific deficits in associative reasoning; ReadBench (Clavié & Brand, 2025) highlights performance degradation in multi-page document understanding; and PixelWorld (Lyu et al., 2025) shows that symbol manipulation capabilities deteriorate in pixel space. Our work accepts this perceptual efficacy as a premise and targets the downstream failure of reasoning activation.

**Cross-Modal Alignment.** Our approach connects to the literature on consistency regularization and cross-view alignment. While early works focused on aligning static feature representations (e.g., CLIP-style contrastive learning (Radford et al., 2021)), recent efforts have shifted towards aligning output distributions. Notably, Wu et al. (Wu et al., 2025) address the text-dependency bias in VLMs by enforcing consistency between responses generated from full-context inputs and those from vision-centric (text-masked) inputs. Although their objective shares mathematical similarities with ours (KL minimization), the problem setting is fundamentally different: they aim to correct *modality imbalance* in mixed-input VQA (preventing the model from ignoring the image), whereas we address *transfer failure* in pure-visual VTC (enabling the model to reason over the image). Concurrently, Vision-SR1 (Li et al., 2025) decomposes VLM reasoning into visual perception and language reasoning stages, using the model's own generated perception to compute self-reward for RL training. The shared motivation is to move beyond sparse final-answer supervision; however, Vision-SR1 targets reward design for standard VQA, whereas CoRe enforces step-wise distribution matching between two semantically equivalent *views* of the same input, leveraging the unique information equivalence of VTC. Our framework treats the text-conditioned policy as a dynamic anchor and applies process-level synchronization (Kim & Rush, 2016) to regularize the visual reasoning trajectory.

**Text-Space and Latent-Space Compression Alternatives.** VTC is not the only route to long-context compression. Token-space methods such as Context Cascade Compression (Liu & Qiu, 2025) and parse-tree guided prompt compression (Mao et al., 2026) operate directly on text representations, pruning or abstracting tokens before they enter

the decoder. These approaches are complementary to ours: they reduce context length without leaving the text modality, whereas VTC converts tokens to pixels and unlocks the vision encoder's ability to absorb information at a different granularity. We do not claim that VTC universally dominates these alternatives; rather, given that VTC provides real system benefits (e.g., 3–4× compression with high OCR fidelity) and is actively being explored in practice, our contribution is orthogonal: *how to make reasoning over visually compressed prompts substantially more reliable*.

**Post-Training for Reasoning.** Enhancing reasoning in LLMs typically relies on Supervised Fine-Tuning (SFT) with Chain-of-Thought data (Wei et al., 2022) and Reinforcement Learning with Verifiable Rewards (RLVR) (Tu et al., 2025). Here RLVR refers to reinforcement learning setups where rewards are computed by automatic, *verifiable* checkers (e.g., exact-match, unit tests, symbolic/math verifiers), enabling large-scale optimization from outcome-level signals (Tu et al., 2025). Algorithms such as PPO (Schulman et al., 2017) and Group Relative Policy Optimization (GRPO) (Shao et al., 2024) have proven effective in math and coding domains by optimizing against outcome correctness, and have been used in large-scale reasoning systems trained primarily from verifiable signals (Guo et al., 2025). However, directly applying these recipes to the visual modality often yields marginal gains due to the exploration difficulty caused by weak initial visual reasoning policies. CoRe complements these paradigms by introducing a dense, token-level supervision signal derived from the model's own textual capabilities.

## 5. Limitations and Future Work

**Dependence on a stronger text anchor.** CoRe is a transfer mechanism, not a method for creating reasoning knowledge from scratch. If the text-conditioned policy cannot solve a task, CoRe cannot teach the visual branch capabilities that are absent in the anchor. The upper bound of the visual performance is strictly limited by the text-view oracle.

**High-fidelity VTC regime only.** Our main experiments intentionally stay in a high-fidelity compression regime with CER below 1%. This isolates reasoning transfer from OCR failure, but it also means that we do not yet characterize the extreme-compression frontier where the renderer itself becomes the dominant bottleneck. Systematically sweeping the compression–accuracy frontier is an important direction for future work.

**Text-heavy evaluation.** The current benchmark suite emphasizes math, tables, and long documents. These are natural stress tests for VTC, but they do not cover settings where non-text visual semantics (e.g., photographs, diagrams, or charts) interact deeply with language. We therefore do not claim that the current results imply unchanged performance on natural VQA or broader multimodal perception tasks, and leave such evaluation to future work.

**Fixed rendering configuration.** We deliberately keep the renderer fixed to make the comparisons interpretable, but in practice, the optimal rendering strategy may vary depending on the document type (e.g., dense tables vs. sparse narratives). Whether CoRe could benefit from dynamic or learnable rendering strategies is an open question.

**Training overhead.** The lockstep alignment introduces computational overhead during training, requiring forward passes through both the text and visual branches. Although the text branch is computationally cheaper (fewer tokens than visual patches) and requires no backward pass due to the stop-gradient operator, this still increases the training cost relative to standard visual-only fine-tuning.

## 6. Conclusion

In this work, we formalized the phenomenon of *post-training transfer failure* in Vision-Text Compression (VTC), identifying that standard visual post-training fails to unlock the reasoning potential of VLMs despite accurate perception. To bridge this gap, we proposed **CoRe**, a plug-in consistency framework that aligns the visual reasoning process with the stronger textual reasoning process via lockstep KL divergence. By integrating CoRe into both SFT and RLVR pipelines, we demonstrated that high-level reasoning patterns—not just final answers—can be successfully transferred across modalities. Theoretical analysis supports our method by relating its training objective to an upper bound on visual reasoning error. Empirically, CoRe consistently improves visual-prompt performance across mathematical, memory, and retrieval benchmarks, recovering a substantial portion of the performance lost during compression in the high-fidelity VTC regime studied here.

## Impact Statement

This research advances the efficiency of multimodal AI systems by making visual context compression a viable alternative to computationally expensive long-context attention. By enabling models to reason robustly over compressed visual inputs, CoRe contributes to reducing the inference energy costs and latency associated with processing massive documents, potentially democratizing access to long-context capabilities on resource-constrained devices. However, we acknowledge that robust visual text understanding could be misused to bypass text-based safety filters via "jailbreaking via images" (Xu et al., 2024; Cao et al., 2026b). We

therefore encourage the deployment of multimodal safety guardrails that apply equally rigorous checks to visual inputs as they do to textual content, especially in agentic settings (Cao et al., 2026a).

## Acknowledgements

This work is supported by the National Key Research and Development Program of China (Grant No. 2024YFE0105400). Haobo Wang is also supported by the Ningbo Yongjiang Talent Introduction Programme (No. 2023A-399-G). In addition, this work is supported by Ant Group Research Intern Program. We would like to thank all the anonymous reviewers for their insightful comments.

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

# A. Implementation Details

## A.1. Computational Infrastructure

All experiments in this work were conducted on a high-performance computing cluster. Specifically, we utilized **two compute nodes**, each equipped with **8 × NVIDIA A100 GPUs (80GB VRAM)**. The nodes are interconnected via InfiniBand (IB) to support efficient multi-node distributed training. To optimize training efficiency and memory usage, we implemented our training pipeline using the DeepSpeed library with ZeRO-3 optimization stage. We also leveraged FlashAttention to accelerate the attention computation for long sequences.

## A.2. Training Hyperparameters

We provide the detailed hyperparameters used for both the Supervised Fine-Tuning (SFT) and Reinforcement Learning (RLVR/GRPO) phases in Table 5. We performed a light grid search on the learning rate and batch size during the preliminary experiments on the text modality and fixed these parameters for the visual alignment experiments to ensure fair comparison.

*Table 5.* Hyperparameters for CoRe training across different phases.

| Hyperparameter | SFT Phase | RL Phase (GRPO) |
|---|---|---|
| Base Model | Qwen3-VL-8B-Instruct | Qwen3-VL-8B-Instruct |
| Global Batch Size | 128 | 64 |
| Learning Rate | $1 \times 10^{-5}$ | $5 \times 10^{-6}$ |
| LR Scheduler | Cosine Decay | Cosine Decay |
| Warmup Ratio | 0.03 | 0.05 |
| Weight Decay | 0.01 | 0.01 |
| Max Sequence Length | 4096 | 4096 |
| KL Coefficient ($\beta_{KL}$) | - | 0.01 |
| Alignment Weight ($\lambda$) | 0.1 | 0.1 |
| Rollout Sampling ($K$) | - | 8 |
| Gradient Accumulation | 4 | 8 |
| Optimizer | AdamW | AdamW |
| Precision | BF16 | BF16 |

## A.3. Cross-Model Validation

To verify that the transfer-failure pattern is not unique to a single checkpoint, we repeated a small-scale validation on **Qwen2.5-VL-7B** (Table 6). The qualitative pattern matches the main model: visual-side post-training helps over zero-shot prompting, but remains substantially weaker than matched text-side post-training, confirming a stable empirical pattern across model families.

*Table 6.* Cross-model validation on Qwen2.5-VL-7B. Each cell reports visual / text accuracy under matched supervision.

| Setting | MATH-500 | GSM8K | AMC |
|---|---|---|---|
| Zero-shot | 16.60 / 53.80 | 22.49 / 86.74 | 4.82 / 30.12 |
| SFT (1 epoch) | 26.00 / 45.60 | 50.25 / 76.43 | 9.64 / 21.69 |

# B. VTC Rendering Configuration

## B.1. Configuration Search Protocol

The efficacy of Vision-Text Compression (VTC) relies heavily on the rendering configuration $\rho$. An overly dense rendering yields high compression but risks OCR failures, while a sparse rendering inflates the visual token budget. Inspired by the methodology in Glyph (Cheng et al., 2025), we conducted a systematic search to identify the optimal rendering parameters for our base model, Qwen3-VL-8B-Instruct. We specifically optimized for the **Compression-Recall Frontier**: maximizing the number of characters per visual token while maintaining a Character Error Rate (CER) below 1% on the evaluation set.

Following the search space defined in Glyph (refer to Table 8 in (Cheng et al., 2025)), we explored a wide range of parameters. Specifically, we tested font families including `Arial`, `Verdana`, `Times New Roman`, `Consolas`, and `Courier New`; font sizes ranging from 8pt to 16pt; resolutions (DPI) from 72 to 200; and line height multipliers from 1.0 to 1.5. We also varied margins (0–20px) and background colors (White, Light Gray, Beige) to test robustness.

Our empirical results indicate that Qwen3-VL is robust to font variations but sensitive to information density (DPI/Font Size ratio). We identified that a configuration yielding efficient compression without sacrificing legibility offers the best trade-off. This configuration aligns closely with the findings reported in prior work, verifying the generalizability of their rendering protocol.

## B.2. Final Specification

Based on the search above, we fixed the optimal rendering parameters for all experiments in the main paper. We utilize a standard A4 page size ($595 \times 842$ points) rendered at **72 DPI** with 10-pixel margins on all sides. Text is rendered using a 9pt **Verdana** (Sans-serif) font with a tight 10pt line height, left-aligned in black (#000000) on a white (#FFFFFF) background. Advanced settings include auto-cropping width and the last page to minimize whitespace, a horizontal scale of 1.0, and zero border width or padding. Under this configuration, a typical text prompt is compressed into a highly efficient visual representation, achieving a substantial reduction in token usage compared to the raw text input.

## B.3. Compression Statistics

Under the configuration above, the renderer is selected with a CER $< 1\%$ constraint, so the primary bottleneck is reasoning transfer rather than text recognition. The full per-domain compression statistics are shown in Table 7.

*Table 7.* Compression statistics under the fixed rendering configuration. All domains achieve multi-fold compression while maintaining near-perfect OCR accuracy.

| Domain | Compression Range | CER |
|---|---|---|
| Math | $2.97\times$–$3.27\times$ | $< 1\%$ |
| Tabular QA | $2.49\times$–$4.26\times$ | $< 1\%$ |
| Long-context QA | $3.31\times$–$3.64\times$ | $< 1\%$ |

These numbers confirm that the setting is not a marginal compression regime: we already achieve substantial compression across all three domains while keeping OCR fidelity in the high-accuracy region.

## C. Prompt Templates and Examples

To ensure rigor, we employ a strict template for converting textual problems into their VTC counterparts.

### C.1. Input Format Construction

**Textual Prompt** ($P_t$)**.** We use the standard chat template of the base model:

```
<|im_start|>user
{Problem Text}
<|im_end|>
<|im_start|>assistant
```

**Visual Prompt** ($P_v$)**.** The text content is rendered into an image $I_{\text{render}}$. The input sequence is constructed by prepending the image tokens with a specific visual instruction:

```
<|im_start|>user
<|vision_start|>{I_render}<|vision_end|>
Please solve the task shown in
the following images.
<|im_end|>
<|im_start|>assistant
```

### C.2. Concrete Examples

Table 8 illustrates paired inputs for a mathematical reasoning task derived from our training set. Note that the visual prompt contains the exact same semantic information (including the system instructions and the question) as the text prompt but is processed purely through the vision encoder.

*Table 8.* Comparison of paired Textual and Visual Prompts used in CoRe. The example shows a mathematical reasoning task with a specific Chain-of-Thought system prompt.

| **Textual Prompt** ($P_t$) | **Visual Prompt** ($P_v$) |
|---|---|
| **Input Tokens:**

Your task is to follow a systematic, thorough reasoning process before providing the final solution. This involves analyzing, summarizing, exploring, reassessing, and refining your thought process through multiple iterations. Structure your response into two sections: Thought and Solution. In the Thought section, present your reasoning using the format: "`<think>` {thoughts} `</think>`. Each thought should include detailed analysis, brainstorming, verification, and refinement of ideas. After "`</think>`," in the Solution section, provide the final, logical, and accurate answer, clearly derived from the exploration in the Thought section. If applicable, include the answer in \boxed{} for closed-form results like multiple choices or mathematical solutions. Here is the question:
A robe takes 2 bolts of blue fiber and half that much white fiber. How many bolts in total does it take? | **Input Tokens:**

Your task is to follow a systematic, thorough reasoning process before providing the final solution. This involves analyzing, summarizing, exploring, reassessing, and refining your thought process through multiple iterations. Structure your response into two sections: Thought and Solution. In the Thought section, present your reasoning using the format: "`<think>` {thoughts} `</think>`". Each thought should include detailed analysis, brainstorming, verification, and refinement of ideas. After "`</think>`," in the Solution section, provide the final, logical, and accurate answer, clearly derived from the exploration in the Thought section. If applicable, include the answer in \boxed{} for closed-form results like multiple choices or mathematical solutions. Here is the question:
A robe takes 2 bolts of blue fiber and half that much white fiber. How many bolts in total does it take?

**Instruction:** Please solve the task shown in the following images. |
| **Model Response (Target):**

`<think>` The user wants to know the total number of bolts of fiber needed for a robe.
1. **Analyze the given information:** Blue fiber needed: 2 bolts. White fiber needed: "half that much" (referring to the blue fiber).
2. **Calculate the amount of white fiber:** Half of 2 bolts = $2/2 = 1$ bolt.
3. **Calculate the total amount:** Total = Blue fiber + White fiber. Total = $2 + 1 = 3$ bolts.
4. **Final Verification:** Blue: 2. White: 1 (half of 2). Total: 3. The logic holds.
`</think>`
The robe takes 2 bolts of blue fiber. The amount of white fiber needed is half of the blue fiber, which is:

$$2 \div 2 = 1 \text{ bolt}$$

To find the total, we add the bolts of blue and white fiber together:

$$2 + 1 = 3 \text{ bolts}$$

$\boxed{3}$ | **Model Response (Expected):**

(We aim to align this output distribution to match the left column via CoRe)
`<think>` The user wants to know the total number of bolts... |

