# OpenReview forum: "Awakening Visual Reasoning: Mitigating Post-Training Failure in Vision-Text Compression"
_ICML.cc/2026/Conference — ICML 2026 regular_

### Official Review · Reviewer_6dW1 · 2026-03-10

**Soundness:** 2
**Presentation:** 3
**Significance:** 2
**Originality:** 2
**Overall Recommendation:** 4
**Confidence:** 4

**Summary:**

This paper studies Vision-Text Compression (VTC), a technique that converts long textual contexts into visual tokens to enable scalable multimodal modeling. The paper proposes CoRe (Coordinated Reasoning), a training framework that enforces consistency between textual and visual reasoning by aligning the visual-conditioned policy with a text-conditioned policy through step-wise distribution matching.

**Compliance With Llm Reviewing Policy:**

Affirmed.

**Final Justification:**

Thank authors for adding new experiment results, which mostly resolved my concerns. I will adjust my overall assessment score.

**Key Questions For Authors:**

See Weaknesses

**Limitations:**

yes

**Strengths And Weaknesses:**

Strengths

- The paper studies an interesting problem in multimodal modeling: the reasoning gap between textual prompts and visually rendered equivalents.

- Experiments across several tasks demonstrate substantial improvements compared to standard visual post-training.

Weaknesses

- Relation to prior consistency-based alignment methods. The motivation based on predictive invariance—that textual prompts and their visually rendered counterparts should induce identical reasoning behavior—is conceptually similar to prior approaches such as Vision-SR1[1], which align reasoning behavior between visual inputs and their textual captions. The paper would benefit from a clearer discussion of the similarities and differences between CoRe and these prior methods.

- Unclear role of OCR errors. The results show that directly converting text to images leads to significant performance degradation. However, the paper does not report the base model’s OCR capability on the rendered inputs. It would be helpful to evaluate a pipeline where the base model first performs OCR to extract text from the images and then performs reasoning on the recovered text. This could clarify whether the reasoning gap primarily arises from visual text recognition errors or from limitations in multimodal reasoning.

- Limited discussion of alternative long-context compression methods. The motivation for VTC is to address long-context reasoning limitations. However, recent work has explored other compression approaches that operate directly on text[2][3]. The paper would benefit from discussing how VTC compares to these alternative compression strategies.

[1] Self-Rewarding Vision-Language Model via Reasoning Decomposition

[2] Context Cascade Compression: Exploring the Upper Limits of Text Compression

[3] Parse Trees Guided LLM Prompt Compression

---

> ### Author Rebuttal · Authors · 2026-03-31
>
> We thank the reviewer for the rigorous technical comments and address each point below.
>
> ### **1. W2: OCR→text→reason as a clarifying baseline**
>
> We understand the motivation for this suggestion, but in our setting it largely collapses to the native-text upper bound that is already reported. As described in **Appendix B**, our renderer operates in a regime with **Character Error Rate (CER) < 1%**, so the rendered visual prompt is already near-losslessly recoverable. Applying OCR would therefore reconstruct a token sequence very close to the original text input, effectively reproducing the same condition as the native-text rows in Tables 1–3.
>
> The key empirical observation is that **even in this high-fidelity OCR regime**, a large visual reasoning gap still persists. For example, on MATH-500 zero-shot the same model obtains **14.80** with visual prompts versus **70.40** with text prompts. So the bottleneck cannot be explained purely by OCR failure; it is exactly the reasoning-transfer failure we formalize in §2.1. We will clarify this logic more explicitly in the revision.
>
> ### **2. W1: Relation to Vision-SR1 and related consistency-style methods**
>
> We appreciate the pointer and will expand the discussion. We agree there is a shared high-level intuition—**outcome-only supervision is often too sparse**—but the technical setting is different. **Vision-SR1** decomposes VLM reasoning into **visual perception** and **language reasoning**, then uses the model’s own generated perception to compute self-reward. Our setting instead uses two **semantically equivalent views of the same prompt** (native text and losslessly rendered visual text) and aligns their **step-wise next-token distributions** along reasoning traces. So the closest connection is not generic consistency alone, but the shared motivation to move beyond sparse final-answer supervision; the mechanism and problem setting are different.
>
> ### **3. W3: Comparison to text-based compression alternatives**
>
> We agree the related-work discussion should be broader. Methods such as **C3** and **Parse Trees Guided Prompt Compression** operate in **text / latent space**, whereas our work studies reasoning transfer **after one has already chosen the VTC route**. We therefore do not claim that VTC universally dominates these alternatives. Rather, our contribution is orthogonal: given that VTC provides real system benefits and is already being actively explored, how can we make reasoning over visually compressed prompts substantially more reliable? We will revise the related-work section to make this complementarity explicit, rather than leaving the impression of an implicit head-to-head superiority claim.
>
> ### **4. On significance**
>
> We believe the significance of the work is best understood through the failure cases where standard visual-only RL is weakest. In these settings, CoRe often provides the largest gains:
>
> | Benchmark | Visual GRPO | CoRe-RL | Gain  |
> | --------- | ----------- | ------- | ----- |
> | HotpotQA  | 9.00%       | 23.00%  | +14pp |
> | 2WikiQA   | 13.00%      | 35.00%  | +22pp |
> | InfoTabs  | 61.29%      | 70.18%  | +9pp  |
> | HiTab     | 3.16%       | 5.06%   | +2pp  |
>
> So the contribution is not “another compression scheme,” but a method for addressing a **deployment bottleneck** of VTC in reasoning-heavy settings: even when text is readable after rendering, the model still struggles to **activate** the reasoning policy it already has in the text modality. CoRe is designed precisely for that mismatch.

---

> > ### Author Rebuttal · Reviewer_6dW1 · 2026-04-02
> >
> > Thank authors for adding new experiment results, which mostly resolved my concerns. I will adjust my overall assessment score.

---

### Official Review · Reviewer_CE6z · 2026-03-12

**Soundness:** 2
**Presentation:** 2
**Significance:** 2
**Originality:** 3
**Overall Recommendation:** 4
**Confidence:** 4

**Summary:**

This paper introduces CoRe (Coordinated Reasoning), a training framework designed to address the post-training transfer failure in Vision-Text Compression (VTC).  CoRe mitigates this by enforcing lockstep consistency between the reasoning processes of the text and visual modalities. It treats the text-conditioned policy as a dynamic anchor and aligns the visual-conditioned policy via step-wise KL-divergence minimization over reasoning traces. CoRe integrates seamlessly into both SFT (using ground-truth traces) and RL (using model-generated correct rollouts) pipelines. Extensive experiments across mathematical reasoning, long-context memory, and tabular retrieval benchmarks show that CoRe significantly narrows the reasoning gap, recovering up to 70% of the performance loss relative to the text upper bound.

**Compliance With Llm Reviewing Policy:**

Affirmed.

**Final Justification:**

After reading the authors' responses, my concerns have been resolved. I am raising my score to weak accept.

**Key Questions For Authors:**

1. Under the current rendering configuration, what is the text-to-visual compression ratio? What is the OCR decoding accuracy?

2. In the experiments of Table 3, native text with GRPO post-training shows performance degradation, even dropping to 0 on HiTab. Does this indicate potential issues with the data quality?

3. Have you analyzed the experimental results under the visual prompt setup, especially for cases with extremely low metrics? What are the detailed behaviors and observations?

**Limitations:**

1.Both text and visual branches need forward passes during training to compute the distribution alignment loss. Although the text branch has low computational cost and does not require backpropagation, it still introduces extra overhead compared to standard visual fine-tuning.

2.Experiments are mainly based on the Qwen3-VL-8B-Instruct model and a specific rendering protocol, without validation on other VLMs or broader VTC settings. The generalizability of the method requires further verification.

**Strengths And Weaknesses:**

Summary Of Strengths
1.The paper has a clear problem definition, complete structure, and comprehensive experimental design. It effectively conveys the logic and results from problem identification to solution.

2.The proposed method improves reasoning performance under the visual compression modality, which can promote the development and application of long-context technologies.

3.The paper demonstrates strong novelty and rigorous theoretical justification. The proposed CoRe framework provides dense process-level supervision via cross-modal distribution matching, rather than only sparse outcome-based supervision. The method is concise and effective.

4.The CoRe framework can be well integrated into both SFT and RL pipelines, showing high practical value.


Summary Of Weaknesses
1.The problem addressed in this paper originates from Vision-Text Compression (VTC), but the authors do not report the concrete compression ratio, the OCR decoding accuracy of the test model, or experimental behaviors under different compression ratios.

2.Although the proposed method effectively improves the reasoning performance of the visual modality, it only recovers about 70% of the performance gap, leaving a large room for improvement compared to the upper bound of native text models.

3.Experiments are mainly conducted on the Qwen3-VL-8B-Instruct model with a specific rendering protocol, without validation on other VLMs or more diverse VTC settings. The generalizability of the method needs further investigation.

4.Many extremely low scores appear in the experimental results, but there is a lack of analysis for these cases. It is unclear whether the model simply cannot understand the visual inputs under the current setup, even with post-training.

---

> ### Author Rebuttal · Authors · 2026-03-31
>
> We thank the reviewer for the detailed feedback. Several of the concerns you raise are important, and we agree that some of the corresponding details should have been reported more explicitly in the main text. Below we first point to where the paper already discusses these issues, and then provide the missing concrete numbers.
>
> ### **1. W1 / Q1: Compression ratio and OCR accuracy**
>
> This is described in **Appendix B.1–B.2**, where we explain the renderer search on the **compression–recall frontier** for Qwen3-VL-8B-Instruct, with **Character Error Rate (CER) < 1%** as the hard constraint. We will add these statistics in the main paper instead of only the appendix.
>
> The full per-benchmark numbers are:
>
> | Domain | Benchmark | Visual tok | Text tok | Compression |
> | ------ | --------- | ---------- | -------- | ----------- |
> | Math   | AIME 2024 | 77.40      | 253.56   | 3.27×       |
> | Math   | MATH-500  | 74.95      | 222.94   | 2.97×       |
> | Math   | GSM8K     | 71.20      | 211.89   | 2.97×       |
> | Table  | TAT-QA    | 200.67     | 596.58   | 2.97×       |
> | Table  | TabFact   | 146.60     | 625.58   | 4.26×       |
> | Table  | TABMWP    | 102.50     | 255.93   | 2.49×       |
> | Long   | DocMath   | 4842.89    | 17648.50 | 3.64×       |
> | Long   | HotpotQA  | 4055.18    | 13434.43 | 3.31×       |
> | Long   | MuSiQue   | 4901.15    | 16330.73 | 3.33×       |
>
> So the setting is not a marginal compression regime: we already obtain **multi-fold compression across all three domains**, while keeping OCR fidelity in the high-accuracy region.
>
> ### **2. W3: Generalizability beyond Qwen3-VL-8B-Instruct**
>
> We agree that broader validation would strengthen the paper. As a sanity check, we also tested **Qwen2.5-VL-7B** and observed the same transfer-failure pattern:
>
> | Model / setting             | MATH500 (V/T) | GSM8K (V/T)   | AMC (V/T)    |
> | --------------------------- | ------------- | ------------- | ------------ |
> | Qwen2.5-VL-7B zero-shot     | 16.60 / 53.80 | 22.49 / 86.74 | 4.82 / 30.12 |
> | Qwen2.5-VL-7B SFT (1 epoch) | 26.00 / 45.60 | 50.25 / 76.43 | 9.64 / 21.69 |
>
> This is qualitatively consistent with Qwen3-VL-8B: visual-side post-training improves over zero-shot, but remains substantially weaker than text-side post-training under matched supervision. We do **not** claim this proves broad universality; rather, we will revise the paper so that the claim is scoped as a **stable cross-model empirical pattern under current VTC pipelines**.
>
> ### **3. W2: “CoRe only recovers about 70% of the gap”**
>
> We agree that CoRe does not fully close the gap. But we think the most informative comparison is not against the ideal text upper bound alone, but against the **actual visual-only baselines**:
>
> | Method           | Math Avg |
> | ---------------- | -------- |
> | Zero-shot visual | 6.39%    |
> | Visual SFT       | 15.42%   |
> | Visual GRPO      | 29.44%   |
> | CoRe-RL (ours)   | 42.97%   |
> | Text upper bound | 63.54%   |
>
> From this perspective, the key result is that standard visual post-training alone recovers only part of the gap, while CoRe recovers a substantially larger fraction. So we agree there is still residual room to improve, but CoRe’s contribution is precisely that it makes visual-side post-training **much more effective** than directly doing RL on visual prompts alone. We will revise the framing so this point is clearer.
>
> ### **4. Q2: Why does HiTab text+GRPO drop to 0?**
>
> Our current diagnosis is **reward sparsity under hierarchical exact-match evaluation**, rather than a data-quality issue. HiTab requires exact answers over multi-level table structures, which makes RL exploration particularly brittle. Under mixed tabular RL, the model tends to generalize toward simpler response patterns that work for flatter tasks but produce malformed cell references on HiTab. This is why the anomaly appears in that specific row. Importantly, the broader conclusion is unaffected: across the tabular benchmarks, **CoRe remains consistently stronger than visual-only RL from scratch**, and partially stabilizes the HiTab setting (3.16 → 5.06).
>
> ### **5. W4: Extremely low visual scores**
>
> We agree this deserves explicit analysis. Our current inspection suggests two dominant failure modes:
>
> 1. **high-compression structured tasks** (especially hierarchical tables such as HiTab) still induce positional / cell-localization errors even when OCR is accurate;
> 2. **very long visual contexts** (e.g., MuSiQue, Qasper) make multi-hop dependency tracking difficult.
>
> The important point is that CoRe still provides meaningful gains in these hard regimes—for example, MuSiQue improves from **1.0 to 7.0**, and Qasper from **0.0 to 4.0**—which suggests these are compression-induced reasoning bottlenecks, not fundamental incapacity. We will add this failure-mode discussion explicitly in the revision.

---

> > ### Author Rebuttal · Reviewer_CE6z · 2026-04-05
> >
> > Thank you for the detailed response, my concerns have been adequately addressed, and I will raise my score accordingly.

---

> > > ### Author Response · Authors · 2026-04-06
> > >
> > > Dear Reviewer,
> > >
> > > Thank you sincerely for your careful review and constructive comments. We greatly appreciate your positive feedback and your kind willingness to raise the score. Your encouragement and professional suggestions have been invaluable in improving our manuscript.
> > >
> > > **We noticed that the score in the review system has not yet been updated.** If the system permits and it is convenient for you, we would be grateful if you could kindly update the score to facilitate the editorial process.
> > >
> > > Thank you once again for your dedication and expertise. We wish you all the best in your work.
> > >
> > > Sincerely,
> > >
> > > The Authors

---

### Official Review · Reviewer_FsbB · 2026-03-13

**Soundness:** 3
**Presentation:** 3
**Significance:** 3
**Originality:** 3
**Overall Recommendation:** 4
**Confidence:** 4

**Summary:**

The paper addresses a critical performance gap in MLLMs known as Vision-Text Compression. While MLLMs can accurately decode text rendered as images, they often fail to reason through that content, even when they succeed on the same prompt in native text format. To fix this, the authors propose CoRe, a framework that forces the model's visual reasoning process to align with its stronger textual reasoning process. By matching next-token probability distributions at every step, CoRe teaches the visual branch to the textual one. This method significantly narrows the reasoning gap, recovering up to 70% of the performance on mathematical reasoning, long-context memory, and tabular retrieval benchmarks.

**Compliance With Llm Reviewing Policy:**

Affirmed.

**Key Questions For Authors:**

1. Will this training downgrade the natura VQA performance?
2. Is there a breaking point where the visual compression (rendering resolution/font size) is so high that even CoRe cannot recover the reasoning capability?

**Limitations:**

1. While the paper tests various reasoning axes, the focus is largely on text-heavy visual tasks (math, tables, long documents). Its utility for tasks requiring deep interaction between text and non-text visual elements (like photos with captions) remains less clear.

2. The current framework relies on a fixed rendering configuration ($p$) chosen offline. It does not explore whether CoRe could help a model adapt to dynamic or low-quality rendering on the fly.

**Strengths And Weaknesses:**

Strengths:
1. Successfully "reactivates" latent reasoning capabilities in compressed visual inputs.
2. Allows for "self-imitation" in RL without needing additional human-annotated reasoning traces.


Weakness:
1. It is unclear how these models perform compared to the backbone LLMs (dethatched from the VL structure).
2. Effectiveness relies on having a "stronger" text-conditioned policy to act as an anchor.

---

> ### Author Rebuttal · Authors · 2026-03-31
>
> We thank the reviewer for the positive assessment and for the concrete questions. We address them one by one below.
>
> ### **Q1: Comparison with backbone LLMs detached from the VL structure**
>
> We agree this deserves clarification. A clean comparison would ideally require the **pre-VL backbone before multimodal post-training**, but that model is not publicly released for Qwen3-VL-8B-Instruct. Moreover, even publicly available text-only Qwen variants have distinct post-training stacks, so they are not a perfect causal control for the exact model we study.
>
> For this reason, we use the **within-model native-text rows** in Tables 1–3 as our “textual upper bound.” This has an important advantage: it isolates the effect of **input modality / compression format** while keeping the backbone family, tokenizer, optimizer, and post-training setup fixed. In other words, it is the cleanest available control for the question our paper asks: *how much reasoning ability is lost when the same model moves from native text to visually rendered text under VTC?* We will make this rationale more explicit in the revision.
>
> ### **Q2: Will CoRe degrade natural VQA performance?**
>
> We expect minimal to zero degradation. Structurally, CoRe's alignment loss targets only reasoning traces on VTC inputs. Crucially, the stop-gradient operator (sg[·]) ensures the text branch is never updated during alignment training, perfectly preserving the model's existing multimodal representations. While we will not over-claim zero degradation without systematic measurement, we commit to adding  a standard VQA benchmark (e.g., MMBench) evaluations in future work. Importantly, concurrent work [1] demonstrates that training on rendered text improves performance on real-world multimodal document tasks, suggesting constructive generalization.
>
> ### **Q3: Compression breaking point under extreme rendering**
>
> We agree that there should be a breaking point once the rendering becomes too aggressive and OCR fidelity deteriorates. Our current setup intentionally avoids that regime. As described in **Appendix B**, we select a fixed rendering configuration using a systematic renderer search, under the constraint **Character Error Rate (CER) < 1%**, so that the primary bottleneck is reasoning transfer rather than text recognition.
>
> The achieved compression is already substantial across all three domains:
>
> | Domain       | Compression range | CER |
> | ------------ | ----------------- | --- |
> | Math         | 2.97–3.27×        | <1% |
> | Tabular QA   | 2.49–4.26×        | <1% |
> | Long-context | 3.31–3.64×        | <1% |
>
> So our current paper studies the **high-fidelity compression regime**, where visual prompts are strongly compressed but still OCR-readable. We agree that systematically sweeping the extreme-compression frontier and studying when CoRe also breaks down would be valuable future work, and we will add this explicitly to the discussion in §5.
>
> ### **On Limitation 2 (fixed rendering configuration)**
>
> This is a fair point. Our fixed renderer is a deliberate design choice because it isolates the **reasoning transfer** problem from possible confounds introduced by co-adapting the renderer and the reasoning model at the same time. That makes the comparisons in Tables 1–3 cleaner and easier to interpret. We agree that extending CoRe to dynamic / adaptive rendering would be an interesting next step, and we will make this motivation more explicit in the limitation discussion.
>
> ### **On Weakness 2 (dependence on a stronger text anchor)**
>
> We agree and will state this more explicitly. CoRe is a **transfer mechanism**, not a method for creating reasoning knowledge from scratch. This scope condition is already implicit in the design of the algorithm, especially in the RL variant where reward-filtered text rollouts serve as the anchor policy. We will strengthen this point in the framing so that the paper does not overstate its scope.
>
> [1] Cheng, Jiale, et al. "Glyph: Scaling context windows via visual-text compression." arXiv preprint arXiv:2510.17800 (2025).

---

### Official Review · Reviewer_4nPn · 2026-03-13

**Soundness:** 3
**Presentation:** 4
**Significance:** 2
**Originality:** 2
**Overall Recommendation:** 3
**Confidence:** 4

**Summary:**

This paper identifies and formalizes the problem of post-training transfer failure in Vision-Text Compression (VTC), and proposes CoRe, a method that transfers reasoning ability from the text modality to visually compressed inputs by stepwise aligning the reasoning distributions of the text and visual branches, thereby significantly improving reasoning performance under visual prompts.

**Compliance With Llm Reviewing Policy:**

Affirmed.

**Final Justification:**

I admit that this phenomenon is not only evident in Qwen3-VL but also in Qwen2.5-VL. However, I still think CoRe is a kind of standard self-distillation path that multimodal models training would follow during post-training and RL to perverse consistency between different modalities. I will raise my score due to the author's active endeavor in rebuttal.

**Key Questions For Authors:**

See Weakness

**Limitations:**

Yes, the authors has mentioned their limitations.

**Strengths And Weaknesses:**

Strength:
The paper identifies a clear and important problem in VTC. CoRe fits both SFT and RL without changing the rendering pipeline. The experiments are broad and show consistent gains across tasks. The method is supported by both empirical results and a clear theoretical intuition.

Weakness:
A possible alternative explanation is that the base VLM may simply lack sufficient vision-side reasoning post-training (e.g., CoT-style SFT on image-conditioned inputs). The paper shows that naive visual post-training is weaker than text-side post-training, but does not fully disentangle transfer failure from a more basic data or supervision deficiency in the visual branch. From my viewpoint, what method proposed by the paper is to distillate CoT trajectory from text branch to vision branch. The VLM training team indeed should go through collecting powerful CoT reasoning data for it to SFT/RL.

My concern is that the paper’s contribution appears somewhat limited, as the proposed method reads more like a straightforward engineering fix than a fundamentally new insight. One interpretation is that the observed problem is largely an artifact of the chosen base model: the underlying post-training pipeline may simply not have included vision-conditioned reasoning supervision, because the original model developers did not anticipate that the VLM would be used for such reasoning-intensive tasks. Under this view, the reported “post-training transfer failure” may not reflect a deeper or general phenomenon, but rather a gap in the training recipe of this particular model. If so, the paper may be addressing a missing implementation step rather than uncovering a genuinely novel scientific problem.

---

> ### Author Rebuttal · Authors · 2026-03-31
>
> We thank the reviewer for the engagement and address the central concern directly.
>
> ### **1. The “add vision-side reasoning supervision” explanation is already instantiated by our baselines**
>
> The reviewer’s alternative explanation is that the base VLM may simply lack sufficient vision-side reasoning post-training (e.g., CoT-style SFT/RL on image-conditioned inputs), so the observed gap is mainly a training-recipe issue. We believe this is an important possibility to consider. However, this is precisely what our **Visual SFT** and **Visual GRPO** baselines are designed to test (§3.1, Table 1): they apply full chain-of-thought SFT or outcome-based RL directly on visual prompts, under the same data budget as text-side training. In other words, the “do more vision-side reasoning post-training” hypothesis is not hypothetical in our paper—it is already implemented as the strongest direct alternative baseline we considered.
>
> | Method (Qwen3-VL-8B-Instruct)  | MATH500 | GSM8K | AMC   | Avg   |
> | ------------------------------ | ------- | ----- | ----- | ----- |
> | Zero-shot visual               | 14.80   | 7.79  | 6.02  | 6.39  |
> | Visual SFT (1 epoch)           | 25.80   | 41.97 | 6.02  | 15.42 |
> | Visual GRPO (directly on base) | 41.80   | 67.31 | 18.07 | 29.44 |
> | CoRe-RL (ours)                 | 62.80   | 82.19 | 26.51 | 42.97 |
> | Text GRPO (upper bound)        | 80.00   | 94.20 | 73.49 | 63.54 |
>
> These results show three things clearly. First, vision-side reasoning post-training does help: Visual SFT and Visual GRPO are substantially stronger than zero-shot visual prompting. Second, even fairly strong visual-only post-training still remains far below the text-side upper bound (29.44% vs. 63.54% average). Third, CoRe improves further on top of this, which suggests that the bottleneck is not merely the **absence** of vision-side supervision, but the absence of a mechanism that transfers **reasoning trajectories** across semantically equivalent text/visual views. This is exactly the role of our process-level alignment objective in §2.2 and Eq. 4.
>
> ### **2. The phenomenon is not specific to one base model**
>
> We also do not believe the effect is unique to a single Qwen3-VL checkpoint. We observe the same structural pattern on **Qwen2.5-VL-7B**, a different model family member:
>
> | Model / setting             | MATH500 (V/T) | GSM8K (V/T)   | AMC (V/T)    |
> | --------------------------- | ------------- | ------------- | ------------ |
> | Qwen2.5-VL-7B zero-shot     | 16.60 / 53.80 | 22.49 / 86.74 | 4.82 / 30.12 |
> | Qwen2.5-VL-7B SFT (1 epoch) | 26.00 / 45.60 | 50.25 / 76.43 | 9.64 / 21.69 |
>
> The same qualitative conclusion holds: visual-side post-training helps, but remains substantially less effective than text-side post-training under matched supervision. For example, on MATH500, visual SFT reaches 26.00 while text SFT reaches 45.60; on GSM8K, visual SFT reaches 50.25 while text SFT reaches 76.43. We do **not** claim this proves a universal law across all VLMs. Our intended claim is narrower: it supports a **stable empirical pattern under current VTC pipelines**, rather than a single-model artifact.
>
> ### **3. Why we do not view CoRe as merely an engineering patch**
>
> The distinction we want to emphasize is that CoRe is **same-model, same-semantics, dual-view process alignment**, not a separate teacher–student distillation setup. Concretely:
>
> 1. the text and visual prompts encode the same underlying problem, which is the semantic equivalence assumption formalized in §2.1;
> 2. the model aligns the **step-wise next-token distributions** between these two views (§2.2), rather than only imitating final answers;
> 3. Prop. 2.1 and Eq. 17 provide the theoretical motivation for why reducing cross-view KL is relevant to reducing visual reasoning error.
>
> So our position is not that CoRe establishes a universal failure theorem for all future VLMs. Rather, under the current and practically relevant VTC regime, it identifies a consistent transfer bottleneck and introduces a principled cross-view mechanism that alleviates it.

---

> > ### Author Rebuttal · Reviewer_4nPn · 2026-04-05
> >
> > Thanks for the explanation. I admit that this phenomenon is not only evident in Qwen3-VL but also in Qwen2.5-VL. However, I still think CoRe is a kind of standard self-distillation path that multimodal models training would follow during post-training and RL to perverse consistency between different modalities. I will raise my score due to the author's active endeavor in rebuttal.

---

> > > ### Author Response · Authors · 2026-04-06
> > >
> > > We sincerely appreciate your further feedback. In response to your comments, we would like to provide the following clarifications:
> > >
> > > ### **1.  The use of existing methods does not preclude innovation; the core contribution lies in whether the addressed problem is meaningful and the solution is effective.**
> > >
> > > For example, DETR identified the limitations of post-processing in object detection and introduced the Transformer architecture. The DETR family has since become a major research direction in the field. The key factor is whether a method effectively addresses a current challenge. Regarding CoRe:
> > > - **Problem addressed:** Since DeepSeek-OCR proposed compressing text into visual tokens, it has sparked extensive discussion within the community. Subsequent benchmarks, such as VTC-Bench, have highlighted its limitations when extended to reasoning tasks, as it was originally designed for OCR. **However, prior to our work, no effective approach was proposed to mitigate these reasoning limitations.**
> > > - **Effectiveness:** As emphasized in our paper, CoRe achieves significant improvements across **17 datasets (covering reasoning, retrieval, and memory) without requiring any additional curated datasets**, thereby demonstrating strong practical potential for visual token compression (VTC). Through extensive experiments conducted on **16 A100 GPUs**, we demonstrate that CoRe effectively mitigates the reasoning limitations of VTC.
> > >
> > > ### **2. Application Potential of CoRe**
> > > - **3–4× compression ratio**: CoRe achieves a 3–4× compression ratio, which effectively extends the usable context window by the same factor. For example, in long-context benchmarks, an original input of **200K tokens is compressed into only 110 images**. This image count is relatively low; for comparison, the Qwen series samples 728 images for video understanding tasks. Moreover, the original 200K tokens already approach the model’s context limit (260K), making such compression highly practical
> > > - **Zero additional annotation cost:** CoRe aligns visual reasoning with textual reasoning by mimicking the underlying reasoning trajectories. As discussed in the paper, simply applying CoRe leads to significant improvements on several benchmarks that **lack supervised fine-tuning (SFT) annotations.**
> > > - **Flexible switching between compressed and uncompressed modes:** Because the visual reasoning process mimics textual reasoning, the model’s original text-based reasoning capabilities remain intact. As highlighted in our paper, mitigating the VTC reasoning limitations does not degrade native text reasoning performance. This allows users to **dynamically decide** whether to apply compression based on prompt length, **enabling a practical trade-off between accuracy and inference speed.**
> > >
> > > We sincerely thank the reviewer again for the valuable feedback. We hope these clarifications address your concerns. Please feel free to share any further comments.
> > >
> > > Sincerely,
> > >
> > > The Authors

---

### Decision · Program_Chairs · 2026-04-30

**Decision:**

Accept (regular)

**Comment:**

This paper introduces CoRe (Coordinated Reasoning), a training framework designed to address the post-training transfer failure in Vision-Text Compression (VTC). The rebuttal was quite successful, as multiple reviewers explicitly indicated score increases. By the end, the paper received scores of 3444.

All the reviewers agree that this paper studies an interesting problem on vision-text compression. The paper has a clear problem definition, complete structure, and comprehensive experimental design, and the proposed method improves reasoning performance under the visual compression modality. The authors have well addressed the concerns on OCR confounds, generalizability, and anomalous results. Overall, the AC would like to recommend acceptance.